# Differential *ESR1* Promoter Methylation in the Peripheral Blood—Findings from the Women 40+ Healthy Aging Study

**DOI:** 10.3390/ijms21103654

**Published:** 2020-05-21

**Authors:** Elena S. Gardini, Gary G. Chen, Serena Fiacco, Laura Mernone, Jasmine Willi, Gustavo Turecki, Ulrike Ehlert

**Affiliations:** 1Clinical Psychology and Psychotherapy, Department of Psychology, University of Zurich, 8050 Zurich, Switzerland; e.gardini@psychologie.uzh.ch (E.S.G.); s.fiacco@psychologie.uzh.ch (S.F.); l.mernone@psychologie.uzh.ch (L.M.); j.willi@psychologie.uzh.ch (J.W.); 2University Research Priority Program (URPP) Dynamics of Healthy Aging, University of Zurich, 8050 Zurich, Switzerland; 3Douglas Hospital Research Center, McGill University, Montreal, QC H4H 1R3, Canada; gang.chen@douglas.mcgill.ca (G.G.C.); gustavo.turecki@mcgill.ca (G.T.)

**Keywords:** *ESR1* promoter, DNA methylation, CpGI shore, estradiol, women healthy aging

## Abstract

Background Estrogen receptor α (ERα) contributes to maintaining biological processes preserving health during aging. DNA methylation changes of ERα gene (*ESR1*) were established as playing a direct role in the regulation of ERα levels. In this study, we hypothesized decreased DNA methylation of *ESR1* associated with postmenopause, lower estradiol (E2) levels, and increased age among healthy middle-aged and older women. Methods We assessed DNA methylation of *ESR1* promoter region from dried blood spots (DBSs) and E2 from saliva samples in 130 healthy women aged 40–73 years. Results We found that postmenopause and lower E2 levels were associated with lower DNA methylation of a distal regulatory region, but not with DNA methylation of proximal promoters. Conclusion Our results indicate that decreased methylation of *ESR1* cytosine-phosphate-guanine island (CpGI) shore may be associated with conditions of lower E2 in older healthy women.

## 1. Introduction

Human life expectancy has been growing at a rapid rate [1], but this prolongation of life has not been accompanied by a proportional increase in the quality of life [2]. Indeed, along with life expectancy, the incidence of age-related disabilities and comorbidities is also increasing [2]. Women are more likely to survive into older ages than men and are therefore also exposed to a higher risk of age-related disabilities [3]. As such, it is becoming increasingly important to identify factors which influence the physical aging process and extend healthy aging, especially in women [4].

Levels of ovarian hormones have been proposed as important factors influencing health among older women [5,6]. Estradiol (E2), the most potent form of estrogen, has important biological functions in addition to those associated with reproduction. These include regulatory functions of the cardiovascular system, central nervous system, skeletal homeostasis, and lipid and carbohydrate metabolism [7]. In premenopausal women, the ovaries are the principal source of E2 [8]. However, with the onset of menopause, the ovaries’ production of E2 progressively ceases [9]. Therefore, E2 levels decrease as women reach perimenopause, and enter into a low, steady level as women reach postmenopause [10]. Declining E2 levels during perimenopause and low E2 levels during postmenopause appear to contribute to the increased incidence of diseases in older women, such as metabolic diseases [11,12], cognitive decline [13], mental disorders [14,15], osteoporosis, and cardiovascular diseases [16,17]. However, evidence indicates that low levels of estrogen receptors (ERs) may also play a role in the exacerbation of age-related diseases [18,19,20,21]. 

ERs mediate the effects of estrogen through genomic and non-genomic mechanisms across a wide range of cells and tissues [22,23]. Genomic effects, which occur over several hours, are mediated by ERα and ERβ, also known as classical ERs [23]. Following activation by estrogen binding, these ERs translocate from the cell cytoplasm to the cell nucleus, where they contribute to the transcriptional activity of an important number of downstream genes [24,25]. On the cell plasma membrane, ERα, ERβ, and the more recently described G protein-coupled estrogen receptor (GPER) mediate rapid, non-genomic estrogenic effects [26]. 

Studies from humans and rodent models indicate that, among the three ERs, ERα may be a key player in preserving health in advanced age. Indeed, ERα contributes to maintain biological functions such as cardiovascular, metabolic, cognitive, hypothalamic, and limbic functions, even under conditions of low estrogen levels [27,28]. This might be due to the higher affinity of ERα with E2, compared to the affinity with E2 of the other ER subtypes [29,30]. Moreover, studies on osteoporosis indicate that ERα, but not ERβ, is essential in promoting bone-protective actions and bone formation [18,31]. Studies have also reported a greater role of ERα compared to ERβ in protecting cardiovascular functions [18]. Furthermore, a decrease in the relative expression of ERα/ERβ, mainly due to a loss of ERα, is associated with cognitive impairments and a loss of E2 responsiveness in advanced age [27,29]. Concerning GPER, its role as a plasma membrane-based ER is controversial, and there is still a lack of evidence that this ER plays a significant role in mediating endogenous estrogen effects in vivo [30]. 

Levels of ERα are regulated, at least in part, by mRNA expression [32]. DNA methylation is a key epigenetic mechanism, which regulates mRNA expression through the binding of methyl groups at cytosines in cytosine-guanine dinucleotides (CpGs) [33]. These DNA modifications are influenced by various internal and external environmental factors and occur without altering the underlying DNA sequence [34]. Generally, hypermethylation of transcriptional regulatory regions is associated with gene silencing, while hypomethylation is associated with gene activation, resulting in increased mRNA expression [35]. DNA methylation changes have been established as playing a direct role in the transcriptional regulation of ERα gene (*ESR1*) [36]. 

Aging is associated with DNA methylation modifications. Jones et al. (2015) distinguished two categories of age-dependent DNA methylation changes, illustrated by the concepts of “epigenetic drift” and “epigenetic clock” [37]. “Epigenetic drift” refers to modifications that occur due to the loss of regulatory control of DNA methylation mechanisms, and result in increased variability of DNA methylation across aging individuals. By contrast, “epigenetic clock” refers to modifications leading to common DNA methylation changes across aging individuals. Among the common methylation modifications, some may constitute beneficial adaptive changes [37,38,39,40]. These adaptive changes may be the product of natural selection [41,42]. However, associations between DNA methylation and adaptive evolution have not been clearly elucidated. Indeed, DNA methylation marks contribute to adaptive phenotypic variation, but, in mammals, they are erased during early development, following fertilization [41]. Among other hypotheses, beneficial environmentally induced methylation profiles (i.e., changes promoting reproductive functions and longevity) may be maintained across generations through the selection of genomic mechanisms linked to these methylation profiles [41,42].

A recent epigenome-wide analysis may constitute an example of beneficial age-dependent DNA methylation changes [39]. Indeed, age-associated DNA hypomethylation of distal regulatory elements (enhancers) was related to the upregulation of genes essential for cell identity and function. As a consequence, these DNA methylation changes promoted better β-cell function in older mice, suggesting that adaptive responses through DNA methylation changes may occur during aging [39,40]. Regarding *ESR1*, recent evidence suggests that methylation of its promoter may be modifiable across the life span, acting as a regulatory mechanism for ERα expression [32,43,44]. For instance, Ianov et al. (2017) showed that altered methylation of specific CpGs of *ESR1* promoter was associated with age, ovariectomy, and ERα expression in the hippocampus of female rats [32]. To the best of our knowledge, no study has yet investigated potential *ESR1* methylation patterns associated with women’s aging. Changes of *ESR1* promoter methylation may contribute to maintaining health among older women. Therefore, in this study, we hypothesized that lower levels of methylation at three DNA regions of the *ESR1* promoter (proximal promoter A and B, and CpG island shore) would be associated with postmenopause, lower E2 levels, and increased age among healthy women. Our results indicate alterations of *ESR1* cytosine-phosphate-guanine island (CpGI) shore methylation in aging women, and may provide new insights for further investigations in the field of the female health span.

## 2. Results

### 2.1. Description of Demographic and Biological Measures 

The final study population comprised 130 women aged 40–73 years. All women were Caucasian, most were originally from Switzerland (89%) and the remaining part (11%) from the neighboring German-speaking countries, Germany, Austria, and Liechtenstein. Of the total sample, 39.2% (*n* = 51) were premenopausal, 12.3% (*n* = 16) perimenopausal, and 48.5% (*n* = 63) postmenopausal. The majority of the women were married or living together with their partner (68.5%) and had a college/university degree or vocational education (79.2%). Table 1 presents the descriptive statistics of all variables used in this study, in the overall sample and according to menopausal groups. 

### 2.2. ESR1 Promoter Methylation and Menopausal Status 

No statistically significant differences between menopausal groups were detected when analyzing mean methylation of CpG island (CpGI) shore (F (2, 48.032) = 2.08, *p* = 0.137), mean methylation of promoter B (F (2, 39.126) = 0.64, *p* = 0.535), and mean methylation of promoter A (F (2, 127) = 1.02, *p* = 0.362). However, the examination of individual CpGs revealed that methylation at CpG9 of CpGI shore differed significantly between the groups (F (2, 43.953) = 5.08, *p* = 0.010), Figure 1A. Post-hoc analyses indicated significantly lower CpG9 methylation in postmenopausal women (71.1% ± 3.35 SE) compared to premenopausal women (83% ± 2.09 SE, *p* < 0.01). In addition, a significant difference in methylation was detected between groups at CpG7 of promoter B (F (2, 61.065) = 4.58, *p* = 0.014). Methylation at CpG7 was higher in postmenopausal women (3% ± 0.68 SE) than in premenopausal women (1.33% ± 0.25 SE, *p* = 0.060) and perimenopausal women (0.77% ± 0.3 SE, *p* = 0.010). The examination of methylation at individual CpGs of promoter A with respect to menopausal groups did not reveal any statistically significant differences.

### 2.3. ESR1 Promoter Methylation, Estradiol Levels, and Age

The results indicated that E2 levels were significantly predictive of CpGI shore mean methylation (*β* = 0.37, t (118) = 2.14, *p* = 0.034 Figure 1B), while age did not make any significant contribution (*β* = −0.04, t (118) = −0.22, *p* = 0.830). The examination of individual CpGs indicated that E2 levels were predictive of methylation at CpG3 (*β* = 0.54, t (118) = 2.39, *p* = 0.018) and CpG9 (*β* = 0.62, t (118) = 2.38, *p* = 0.019). Table 2 presents the effects of E2 levels and age on methylation at individual CpGs. 

Mean methylation of promoter B was not significantly predicted by E2 levels (*β* = −0.03, t (121) = −1.05, *p* = 0.294) or by age (*β* = −0.04, t (121) = −1.58, *p* = 0.117). However, methylation at CpG 12 was significantly positively predicted by age (*β* = 0.05, t (121) = 2.65, *p* < 0.01). 

Mean methylation of promoter A was not significantly predicted either by E2 levels (*β* = −0.04, t (123) = −1.00, *p* = 0.318) or by age (*β* = 0.03, t (123) = 0.72, *p* = 0.471). Likewise, no significant difference in methylation was observed when examining individual CpGs.

### 2.4. CpGI Shore Methylation and Estradiol Levels Among Menopausal Groups

The regression model predicting CpGI shore mean methylation and including the interaction term between E2 and CpGI shore methylation was found to fit the data better than the model without the interaction term (F (2, 115) = 11.22, *p* = 0.004). A significant positive association between methylation and E2 was found in premenopausal (*β* = 0.56, t (113) = 2.71, *p* = 0.008) and in postmenopausal women (*β* = 0.55, t(113) = 2.04, *p* = 0.044), and no difference emerged between these two groups (*p* = 0.987). By contrast, a significant negative association between methylation and E2 was detected in perimenopausal women (*β* = −0.64, t (113) = −2.26, *p* = 0.026). The examination of individual CpGs indicated that the differential association was stronger at CpG 1 (*p* < 0.001). 

## 3. Discussion

In this study, we explored DNA methylation of key regulatory regions of *ESR1* in association with menopausal status, age, and E2 levels in healthy middle-aged and older women. Methylation levels were low at promoters A and B, while intermediate methylation levels were found at CpGI shore. Postmenopause and lower E2 levels were associated with lower methylation of CpGI shore, and the effect of E2 levels was significant also after adjusting for menopausal status and age. This association was stronger at CpG 3 (Illumina probe: cg07746998), CpG 9 (Illumina probe: cg17264271), and CpG 1 (not included among Illumina probes). CpGI shore methylation was positively associated with E2 levels in pre- and postmenopausal women, while this association was negative in perimenopausal women. Postmenopause was associated with increased methylation of promoter B at CpG 7 (Illumina probe: cg22839866), while age was positively associated with methylation of promoter B at CpG 12 (Illumina probe: cg13612689) after adjusting for E2 levels and menopausal status. Concerning the methylation of promoter A, we did not observe an association with menopausal status, E2 levels, or age. 

Our findings on *ESR1* methylation associated with E2 levels and age present similarities with previous research. First, decreased methylation of CpGI shore at the same CpGs region targeted in the present study was associated with increased age and ovariectomy in the hippocampus (region CA1) of female rats, as well as with E2 deprivation in human breast cancer cells non-resistant to hormone therapy [32,44]. Our results add to these findings and suggest similar associations in peripheral blood cells of healthy women. Second, in the study by Tsuboi et al. (2017), decreased methylation of proximal promoters was not found to be associated with E2 deprivation [44]. Similarly, in our study, decreased methylation of promoter A and promoter B was not associated with menopausal status or E2 levels. On the contrary, consistent with studies showing that methylation at proximal promoters increases with aging [45,46], increased methylation of CpG 7 and CpG 12 in promoter B was associated with postmenopause and increased age, respectively. Thus, contrary to CpGI shore, *ESR1* promoters A and B may not lose methylation in conditions of lower E2 levels, such as increased age and postmenopause. As increased methylation of *ESR1* promoter region (including CpGI shore) has been associated with decreased levels of ERα and increased incidence of age-related diseases [45,46,47,48,49], hypomethylation of CpGI shore in older age may represent a health-promoting mechanism. 

The CpGI shore of promoter C assessed in this study has been described as an enhancer (enhancer ID GH06J151804) of targeted promoters, including promoter A [44,50]. Enhancers are regulatory DNA regions that increase gene transcription by influencing the activity of their target promoters [51]. DNA methylation has been shown to regulate the activity of enhancers, with methylation loss contributing to their activation [52,53,54]. Methylation dynamics at enhancers, marked by intermediate levels of methylation, has also been suggested as a mechanism by which the cell responds to environmental influences, including endogenous changes related to aging [39,55]. We found intermediate levels of methylation in the CpGI shore, suggesting enhancer activity of this sequence in blood. Moreover, decreased methylation at enhancers has been found to correlate with better cell function during aging [39]. These findings may further support the idea that decreased CpGI shore methylation is associated with health during aging.

DNA methylation changes that reflect a programmed process are perhaps selected through evolution [56]. There is increasing evidence indicating that the hypomethylation of enhancers may be an example of these programmed DNA methylation changes [56]. For instance, hypomethylation of enhancers has been implicated as a component of the mouse clock [56], and, as mentioned above, has been shown to preserve cell-functions in mouse pancreatic β-cells during aging [39]. Hormonal changes related to menopause have been described as potential determinants of the epigenetic clock [57]. In this study, we described an association between E2 levels and ERα enhancer methylation that may promote healthy aging. In an evolutionary context, this hormone-DNA methylation association would have positive effects on fitness-related traits earlier in life, as the strength of natural selection decreases with age [58]. 

In our sample, the positive association between CpGI shore methylation and E2 levels observed in pre- and postmenopausal women was not verified in perimenopausal women. This observation may be traced to the previously suggested idea that dysregulation of the estrogen signaling and epigenetic alterations of *ESR1* occur during perimenopause [59]. 

DNA methylation of the CpGI shore may influence mRNA expression by regulating the binding of transcription factors sensitive to DNA methylation. For instance, binding sites for transcription factors of the ETS family are identified in highly conserved regions of the CpGI shore, which include CpG 1, CpG 2, CpG 4, CpG 5, CpG 7–9 [44,60]. Furthermore, a STAT5b binding site is found in a region including CpG 3 [61]. Transcription factors of the ETS family and STAT5b are repressed from binding by methylation within their binding sites in the CpGI shore, leading to decreased *ESR1* expression [44,61,62]. This supports the assumption that DNA methylation differences of CpGI shore found in the present study play a role in regulating CpGI shore transcription.

Finally, our results indicate that E2 was the unique predictor of CpGI shore methylation when controlling for age and menopausal status. E2, through ER, has been shown to exert epigenetic influence on various genes in different tissues, including the blood [63,64]. E2 levels may also contribute to regulate *ESR1* CpGI shore methylation. However, the mechanism underlying the potential regulation of CpGI shore methylation by E2 has not yet been elucidated. As discussed by Ianov et al. (2017), the complex E2-ERα may enhance transcription of repressors interacting with methyltransferases, which in turn would add methyl groups at CpGs of CpGI shore [32]. Thus, a feedback mechanism involving ERα, transcription repressors, and methyltransferases may underlie the association between CpGI shore methylation and E2 levels. 

### Strengths and Limitations 

This is the first study to explore associations between *ESR1* promoter methylation and E2 levels in the context of women healthy aging. In addition, during the participants’ recruitment process, strict inclusion and exclusion health criteria were applied. Therefore, the results could not have been biased by major illnesses.

Although there is evidence indicating that increased CpGI shore methylation is associated with decreased *ESR1* expression in various tissues [32,44,47,61], limitations of this study include the lack of assessment of *ESR1* expression. Furthermore, we assessed DNA methylation only in peripheral blood. This prevents the generalization of results, as DNA methylation may be tissue-specific [65]. However, the blood DNA methylation as a proxy of physiological processes in other tissues has been previously demonstrated [66]. In addition, an epigenome-wide analysis showed that DNA methylation in blood was predictive of all-cause mortality in a sample of 9949 older adults aged 50–75 years [67]. Moreover, the *ESR1* promoter methylation in blood has proved useful in the diagnosis of lung and breast cancers [68,69,70]. Furthermore, it should be noted that methylation at *ESR1* CpGI shore in blood has been found to correlate with the *ESR1* CpGI shore methylation in the brain, especially in the superior temporal gyrus (Appendix A) [65]. Nevertheless, for future studies it would be important to explore the association between CpGI shore methylation, E2 levels, and *ESR1* expression in different cell types. Another limitation is the lack of a longitudinal study design. Indeed, longitudinal data would allow the identification of changes of CpGI shore methylation following the individual variations in E2 levels, age, and menopausal status. Moreover, the sample of perimenopausal women was small (*n* = 16) compared to pre- (*n* = 51) and postmenopausal (*n* = 63) women. Therefore, findings regarding the perimenopausal group must be interpreted with caution. At last, in this study, we did not assess data linked to women’s nutritional status, such as B12 and red cell folate, which have been shown to regulate DNA methylation [71].

## 4. Materials and Methods 

### 4.1. Subjects 

Women aged 40–75 years were recruited in the context of the Women 40+ Healthy Aging Study, a larger cross-sectional investigation including healthy middle-aged and older women [72,73,74]. To be included in the study, women had to report good, very good, or excellent health. Women were excluded from the study if they met at least one of the following criteria: acute or chronic somatic disease; acute or chronic mental disorder; psychotherapeutic treatment and use of psychotropic drugs during the last six months; more than two standard units of alcoholic beverages per day; pregnancy in the last six months; menopause due to surgical removal of the ovaries or the uterus; precocious menopause; current use of oral contraceptives or use of hormone therapy; disease of the thyroid gland, pancreas, adrenal gland or ovaries influencing the endocrine system; diabetes, polycystic ovary syndrome (POCS), hirsutism, endometriosis, and hyper- or hypothyroidism. Subjects were divided into three subgroups, with respect to their menopausal status according to the Stages of Reproductive Aging Workshop +10 (STRAW) criteria: (1) premenopausal, if the menstrual cycle was regular, (2) perimenopausal, if the cycle length was variable, with variability among cycles of at least seven days, or if the interval between cycles was > 60 days, and (3) postmenopausal if no bleeding had occurred in at least the last 12 months [75]. All subjects gave their informed consent for inclusion before they participated in the study. The study (BASEC Nr 2016-01591) was conducted in accordance with the Declaration of Helsinki, and approved on 2 December 2016 by the Cantonal Ethics Committee of the canton of Zurich (KEK Zurich, Zurich, Switzerland).

### 4.2. Biological Sampling 

Saliva and peripheral blood samples were collected at 8:00 am under standardized conditions. In pre- and perimenopausal women, sampling was conducted in the early follicular phase, during which E2 levels are low [76]. 

One or two drops of blood were collected from fingertips onto S&S 903 Whatman^®^ paper cards (GE Healthcare, Little Chalfont, Buckinghamshire, UK). Blood spots were dried at room temperature for about 3 h and stored at −20 °C until subsequent DNA extraction. Participants were asked to collect saliva into 2-ml SaliCaps (IBL International GmbH, Hamburg, Germany) using the passive drool method. Saliva samples were stored at −20 °C until biochemical analysis. 

### 4.3. Methylation Analysis 

DNA isolation—We used the dried blood spot (DBS) technology as source of genomic DNA. Regulation of ERα expression by E2 has been demonstrated in the blood [77]. Therefore, it is possible that DNA methylation changes underlying the ERα regulation by E2 occur in the blood. The use of DBS technology has practical implications in terms of tissue accessibility and storage and has previously been successfully used for evaluating cytosine methylation [78,79]. Genomic DNA was extracted from three punches of 3 mm diameter using theQIAamp DNA Investigator Kit (QIAGEN, Hilden, Germany), following the manufacturer’s instructions, and eluted in a final volume of 30 μL of RNase-free water. Qubit (Thermo Fischer Scientific, Waltham, MA, USA) was used to assess the DNA concentration.

Bisulfite conversion—Genomic DNA (41–168 ng) was bisulfite-treated using the EZ 96-DNA methylation-Gold kit (Zymo Research, Irvine, CA, USA). The manufacturer’s instructions recommend using samples containing 0.5–2000 ng of DNA. Bisulfite converted DNA was eluted in 20 μL of RNase-free water and stored at −80 °C until subsequent analysis. 

NGS Library preparation—We analyzed three DNA sequences located in the promoter region of *ESR1*. Two sequences are located in two CpGI, one in proximal promoter A and one in proximal promoter B. The third sequence is a CpGI shore near promoter C, located approximately 2 kbp upstream of promoter A (Figure 2) [44]. Increased methylation at these regulatory DNA sequences has been shown to decrease *ESR1* expression and to be associated with diseases [45,46,47,48,49]. The PCR amplicon library preparation for next-generation bisulfite sequencing was based on the protocol described by Chen et al. (2017) [80]. An initial polymerase chain reaction (PCR) was performed on the bisulfite-treated DNA using the Kapa HIFI Uracil+ master mix (Kapa Biosystems, Wilmington, MA, USA). Bisulfite primers were designed manually or using MethPrimers [81]. Primers contained universal oligonucleotides CS1/CS2 (Fluidigm, San Francisco, CA, USA, Table 3), used for customized NGS sequencing primers. PCR conditions were 95 °C for 3 min, then 40 cycles of 98 °C for 20 s, 54–60 °C for 15 s, 72 °C for 15 s, and a final step with 72 °C for 45 s. PCR amplicon products were purified using E-gels 2% size selection (Thermo Fisher Scientific, Waltham, MA, USA). To verify that primers were specifically amplifying bisulfite converted DNA, positive and negative controls (bisulfite converted DNA and genomic DNA, respectively) were included in the PCR. Then, a second PCR of 10 cycles (Tm 60 °C) was performed for adding the Illumina NGS library flowcell attachment sites and customized single barcode for each individual (Fluidigm, San Francisco, CA, USA). A final purification of the pooled amplicon libraries from each of the three DNA regions (promoter A, B, and CpGI shore) was performed, and final products were quantified using the Agilent 2200 Tape Station instrument and HS DNA 1000 reagents (Agilent Scientific Instruments, Santa Clara, CA, USA). The three DNA sequences were pooled at a final molarity of 2 nM. To increase the diversity of base calling during sequencing we added PhiX spike-in (12%) to the library. The final library was sequenced on the Illumina MiSeq using the V3, 600 cycles kit (300 PE) (Illumina, San Diego, CA, USA).

Interrogation of CpGs in the targeted amplicons—Adaptor sequences and low-quality bases were removed using the default settings of trimmomatic v0.35 (licensed under GPL V3 and available at http://www.usadellab.org/cms/index.php?page=trimmomatic) [83]. Only paired-end sequences > 2 × 20 bp were kept, which were then aligned to the target regions, and counts were extracted using Bismark program (v0.19.0). A customized R script was subsequently used to parse all counts. In accordance with Chen et al. (2017) [80], a minimum threshold of 100× reads was set. The number of total reads across the samples ranged from 119 to 31,598 (M = 6227, SD = 5240) for promoter A, from 202 to 63,844 (M = 3219, SD = 10,946) for promoter B, and from 148 to 48,993 (M = 19,512, SD = 13,972) for CpGI shore. Finally, unconverted CpGs percentage was calculated for each CpG as the number of unconverted reads divided by the total read count. Levels of methylation were consistent with previous data in blood (retrieved from the GSE40279 dataset [84]). Methylation was low in promoters A and B, while the CpGI shore had intermediate levels of methylation (Table 1). Methylation data are openly available in “Dryad” at https://doi.org/10.5061/dryad.gmsbcc2jk.

### 4.4. Estradiol Measurement

E2 levels were determined using the 17beta-Estradiol Saliva Luminescence Immunoassay (IBL International, Hamburg, Germany). Intra- and inter-assay coefficients were below 13.3% and 14.8%, respectively, and the assay’s analytical sensitivity (limit of detection) was 1.1 pmol/L. According to the manufacturer instructions, E2 values range between 2.9 and 28.2 pmol/L during the follicular phase, while in postmenopausal women expected values are lower than 15.7 pmol/L (IBL International, Hamburg, Germany, RE62141/RE62149). E2 levels were in the ranges suggest by IBL (Table 1). The use of salivary E2 can be justified by the fact that salivary E2 strongly correlates with free serum E2 [85], which is the portion available for estrogenic effects [64,86].

### 4.5. Procedure and Statistical Analyses 

We first compared *ESR1* methylation levels of the three targeted DNA regions among menopausal groups, using analysis of variance (ANOVA). If Levene’s test was significant, the Welch statistics were used, followed by the Games–Howell test for post-hoc comparisons. ANOVA was conducted using SPSS (IBM statistic, version 24.0, Armonk, NY, USA: IBM Corp.). 

Next, we assessed the predictive effect of E2 levels and age on *ESR1* methylation using the robust regression approach. This method allowed to put less weight on more extreme values, which did not present compelling reasons justifying their exclusion (i.e., low sequencing coverage). Menopausal status was also included as a covariate in the regression models. In a second step, we added the interaction term between E2 and menopausal status for assessing possible differential associations between *ESR1* methylation and E2 among menopausal groups. The comparison of the different associations was performed using contrast analysis. Robust regression analyses were conducted using the R-Package ‘robustbase’ in R [87], version 3.5.0 [88]. 

All methylation analyses were performed using the mean methylation of the 27, 12, and 9 CpGs in promoter A, promoter B, and CpGI shore, respectively, and methylation levels at individual CpGs. For most analyses using individual CpGs, only significant results were reported. 

One significant result, with an effect size < 0.04 was considered meaningless [89]. All statistical tests were two-tailed and the significance level was set at *p* < 0.05.

## 5. Conclusions

ERα plays an important role in maintaining health during aging. This report indicates that decreased methylation of *ESR1* CpGI shore may be associated with conditions of lower E2 in older healthy women. This might have important clinical implications in the field of women healthy aging. Future research on this topic may consider gene expression analysis, longitudinal cohorts, and cell-specificity.

## Figures and Tables

**Figure 1 ijms-21-03654-f001:**
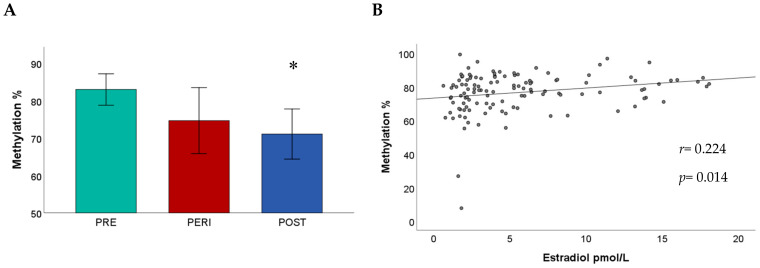
(**A**) Cytosine-phosphate-guanine island (CpGI) shore methylation at CpG9 in menopausal groups. Methylation levels are significantly lower in postmenopausal women compared to premenopausal women. (**B**) The mean methylation of CpGI shore is significantly positively associated with E2 levels (*r* = 0.224, *p* = 0.014). Robust regression was used to put less weight on extreme values. * *p* < 0.05. Abbreviations: PRE = premenopausal; PERI = perimenopausal; POST = postmenopausal.

**Figure 2 ijms-21-03654-f002:**
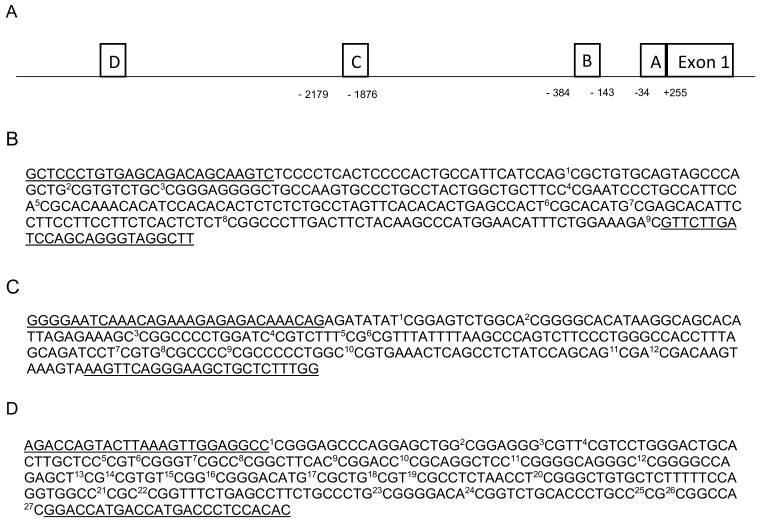
(**A**) Schematic figure of *ESR1* promoter region. (**B**) Assessed DNA sequence (−2179; −876) in CpGI shore near promoter C, including nine CpGs. Methylation at CpGs 1–9 was associated with E2 deprivation in humans [44]. Altered methylation at these CpGs region was associated with ovariectomy and age in the hippocampus of female rats [32]. (**C**) Assessed DNA sequence (−384; −143) in promoter B, including 12 CpGs located in a CpGI. (**D**) Assessed DNA sequence in Promoter A (−34; +255), including 27 CpGs located in a CpGI. Underlined sequences correspond to the primers position.

**Table 1 ijms-21-03654-t001:** Demographic and biological measures.

	All	PRE	PERI	POST
*n*	130	51	16	63
E2 (pmol/L) (mean/SD)	5.6/4.5	8.2/4.7	5.7/4.5	3.7/3.2
Age (y) (mean/median/range)	53.2/52.5/40–73	45.1/45/40–57	51.4/51/47–56	60.2/59/50–73
*ESR1* CpGI shore methylation (%) (mean/SD)	76.7/12	79.3/8.4	78.1/9.2	74.4/14.7
*ESR1* promoter B methylation (%) (mean/SD)	3.5/3.14	3.2/2	3.9/3.3	3.7/3.8
*ESR1* promoter A methylation (%) (mean/SD)	3.9/3.3	3.7/2.5	5/4	3.7/3.7

Note: levels of E2 in pre- and perimenopausal women are measured during the early follicular phase. Abbreviations: PRE = premenopausal; PERI = perimenopausal; POST = postmenopausal; E2 = estradiol; *n* = sample size, *ESR1* = estrogen receptor 1 gene, CpGI = CpG island, SD = standard deviation.

**Table 2 ijms-21-03654-t002:** Effects of estradiol (E2) levels and age on estrogen receptor 1 gene (*ESR1*) CpGI shore methylation at individual CpGs.

	E2	*β*	*p*	Age	*β*	*p*
CpG 1		0.64	0.097		−0.25	0.599
CpG 2		0.12	0.733		−0.42	0.154
CpG 3		0.54	0.018		0.09	0.618
CpG 4		−0.01	0.978		−0.00	0.989
CpG 5		0.42	0.051		0.09	0.661
CpG 6		0.13	0.587		0.14	0.293
CpG 7		0.28	0.100		0.04	0.831
CpG 8		−0.06	0.797		−0.28	0.422
CpG 9		0.65	0.019		−0.33	0.419

Note: methylation at CpG 3 and CpG 9 present significant associations with E2 levels (in bold), while the same association at CpG 1 and CpG 5 presents a trend toward significance. Abbreviations: CpG = cytosine-guanine dinucleotide, E2 = estradiol.

**Table 3 ijms-21-03654-t003:** Primers used for assessing DNA methylation in the promoter regions of *ESR1*.

Target	Forward Primer	Reverse Primer	GRC h37 (hg19)	T (°C)
CpGI shore	ACACTGACGACATGGTTCTACA NNN GTTTTTTGTGAGTAGATAGTAAGTT	TACGGTAGCAGAGACTTGGTCT NNN AAACCTACCCTACTAAATCAAAAAC	chr6: 152126660–152126963	58
Promoter B	ACACTGACGACATGGTTCTACA NNN GGGGAATTAAATAGAAAGAGAGATAAATAG	TACGGTAGCAGAGACTTGGTCT NNN CCAAAAAACAACTTCCCTAAACTT	chr6: 152128433–152128671	60
Promoter A	ACACTGACGACATGGTTCTACA NNN AGATTAGTATTTAAAGTTGGAGGTT	TACGGTAGCAGAGACTTGGTCT NNN ATATAAAAAATCATAATCATAATCC	chr6: 152128780–152129067	54

Note: forward primers include the universal Fluidigm primer sequence CS1 (ACACTGACGACATGGTTCTACA), while reverse primers include the universal Fluidigm primer sequence CS2 (TACGGTAGCAGAGACTTGGTCT). NNN between universal primers CS1/CS2 and bisulfite primers represent randomized nucleotides to molecular diversity generation during sequencing [82]. Abbreviations: CpGI = cytosine-phosphate-guanine island.

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
