# Peer review of "Differential *ESR1* Promoter Methylation in the Peripheral Blood—Findings from the Women 40+ Healthy Aging Study"

_ijms, 2020, doi:10.3390/ijms21103654_

Round 1

Reviewer 1 Report

This is a preliminary description of DNA methylation near the ERS1 gene in peripheral blood of healthy middle aged women grouped according to menopause status and evaluated together with estradiol levels in their saliva. The see some differences that correlate with menopause status or E2 levels. It is not addressed whether this is a characteristic of all genes in the blood, or specific to ESR1. There is no data to indicate whether blood DNA methylation is a proxy for any kind of physiological impact. It is unclear what the rationale of the blood analysis is as it relates to biological importance.

Author Response

Dear Reviewer,

Thank you for revising our manuscript, “Differential ESR1 Promoter Methylation in middle-aged and older women-Findings from the Women 40+ Healthy Aging Study, for publication in the International Journal of Molecular Science, Special Issue “Steroid Metabolism in Human Health and Disease”. We appreciate the interest that you have taken in our report and the constructive criticism you have given. Below, you will find the response to each of the concerns you have raised and the changes made according to your suggestions. We hope that you will be satisfied with our response.

The changes mentioned below have been implemented in the manuscript using the "Track Changes" function in Microsoft Word. Moreover, the manuscript has undergone professional proofreading. Finally, upon editorial request, we looked for an alternative title. Our proposal is: “Differential ESR1 Promoter Methylation in the Peripheral Blood- Findings from the Women 40+ Healthy Aging Study”. The dataset of this study can be accessed through this temporary link: https://datadryad.org/stash/share/9rCnsb4oTu79zdE7aL5DCsoliwyDSariAmI3y0nyuCw.

Comments and suggestions from Reviewer 1 with authors’ responses

This is a preliminary description of DNA methylation near the ERS1 gene in peripheral blood of healthy middle-aged women grouped according to menopause status and evaluated together with estradiol levels in their saliva. They see some differences that correlate with menopause status or E2 levels.

  • It is not addressed whether this is a characteristic of all genes in the blood, or specific to ESR1.

Response: Estrogen, through estrogen receptors, has been shown to exert epigenetic influences on various genes in different tissues, including the blood (Campesi et al., 2012; Zhang and Ho, 2011). This sentence has been added to the manuscript (lns. 16-17, p. 6).

  • There is no data to indicate whether blood DNA methylation is a proxy for any kind of physiological impact.

Response: Although DNA methylation may be tissue-specific (Hannon et al., 2015), the blood DNA methylation as a proxy of physiological processes in other tissues has been previously demonstrated (Smith et al., 2014). In addition, an epigenome-wide analysis showed that DNA methylation in blood was predictive of all-cause mortality in a sample of 9,949 older adults aged 50–75 years (Zhang et al., 2017). Moreover, the ESR1 promoter methylation in blood has proved useful in the diagnosis of lung and breast cancers (Martínez-Galán et al., 2014; Suga et al., 2008). These arguments have been added to the “strengths and limitations” section (lns. 32-37, p. 6).

  • It is unclear what the rationale of the blood analysis is as it relates to biological importance.

Response: Regulation of ERα expression by E2 has been demonstrated in the blood (Kim et al., 2004). Therefore, it is possible that DNA methylation changes underlying the ERα regulation by E2 occur in the blood. This sentence has been added to the “Materials and Methods” section (lns. 32-33, p.7).

We hope that our revision meets your approval. We would be glad to respond to any further questions and comments that you may have.

Yours sincerely,

Ulrike Ehlert

Reviewer 2 Report

This paper investigates methylation levels within the ESR1 promoter in cohort of female >40 years. Although there are some novel and interesting aspects to this work. I have a number of reservations about the work that necessitate addressing before this manuscript being in an acceptable state for publication in an academic journal.

Point 1

Although some of these modifications may be a random by-product of the aging process itself, others may promote beneficial adaptive changes

Are the beneficial adaptive changes a product of natural selection?

Point 2

Are these changes due to ageing or are they a by product of it?

“For instance, an epigenome-wide analysis revealed that age-associated DNA hypomethylation of distal regulatory elements (enhancers) was related to the upregulation of genes essential for cell identity and function”

Line 48. This is a little vague

“All women were Caucasian, and most were Swiss or from neighboring countries”

Line 50 In what was is this relevant to the study?“and had a college/university Int. J. Mol. Sci. 2020, 21, vocational education (79.2%)”

What is relevant to a study of this nature is their nutritional status. For example, it would be worthwhile knowing about intakes of vitamin B12 and folate in a population such as this”

In addition, it would be interesting to know a little about blood marker status and folate metabolism in these subjects e.g.  such as homocysteine levels, B12, red cell folate.

Results

In table one it would be worthwhile including the median age in this cohort

Figure one B why is there no r value on the correlation. Is there a trend. A Persons correlation would be useful

Lines 21-33

“In this study, we explored DNA methylation of key regulatory regions of ESR1 in association with menopausal status, age, and E2 levels in healthy middle-aged and older women. Methylation

levels were low at promoters A and B, while intermediate methylation levels were found at CpGI

shore. Postmenopause and lower E2 levels were associated with lower methylation of CpGI shore,

and the effect of E2 levels was significant also after adjusting for menopausal status and age. This

association was stronger at CpG 3 (Illumina probe: cg07746998), CpG 9 (Illumina probe: cg17264271)

and CpG 1 (not included among Illumina probes). CpGI shore methylation was positively associated

with E2 levels in pre- and postmenopausal women, while this association was negative in

perimenopausal women. Postmenopause was associated with increased methylation of promoter B

30 at CpG 7 (Illumina probe: cg22839866), while age was positively associated with methylation of

promoter B at CpG 12 (Illumina probe: cg13612689) after adjusting for E2 levels and menopausal

status. Concerning methylation of promoter A, we didn’t observe association with menopausal

status, E2 levels or age.”

If we make the assumption that these findings are robust. What can be concluded about the relationship between ageing, the menopause and DNA methylation from an evolutionary context?

Minor changes

Line 39 Please change “to maintain” to “ to maintaining”

Line 40 change “ESR1 promoter” to “ the ESR1 promoter”

Line 43/44 please change  “insight for further investigations in the field of women

44 healthy aging.” to “insights for further investigations in the field of female healthspan”.

Please change

 “Concerning methylation of promoter A, we didn’t observe association with menopausal status, E2 levels or age.”

To

“Concerning the methylation of promoter A, we didn’t observe an association with menopausal status, E2 levels or age.

Please change

“Our findings on ESR1 methylation associated with E2 levels and age present similarities with

35 previous researches.”

To “Our findings on ESR1 methylation associated with E2 levels and age present similarities with 35 previous research”.

This needs to be phrased better

Line 46 “Therefore, our sample was representative of the healthy population and major illnesses couldn’t have biased the results.”

Line 47

Please change

“Although there is evidence indicating that increased CpGI shore methylation is associated to decreased ESR1 expression in various tissues.”

Although there is evidence indicating that increased CpGI shore methylation is associated with decreased ESR1 expression in various tissues

I don’t know if it is a journal requirement but presently where the conclusions are located in the manuscript is not appropriate. Conclusions need to be after the discussion.

Author Response

Dear Reviewer,

Thank you for revising our manuscript, “Differential ESR1 Promoter Methylation in middle-aged and older women-Findings from the Women 40+ Healthy Aging Study, for publication in the International Journal of Molecular Science, Special Issue “Steroid Metabolism in Human Health and Disease”. We appreciate the interest that you have taken in our report and the constructive criticism you have given. Below, you will find the response to each of the concerns you have raised and the changes made according to your suggestions. We hope that you will be satisfied with our response.

The changes mentioned below have been implemented in the manuscript using the "Track Changes" function in Microsoft Word. Moreover, the manuscript has undergone professional proofreading. Finally, upon editorial request, we looked for an alternative title. Our proposal is: “Differential ESR1 Promoter Methylation in the Peripheral Blood- Findings from the Women 40+ Healthy Aging Study”. The dataset of this study can be accessed through this temporary link: https://datadryad.org/stash/share/9rCnsb4oTu79zdE7aL5DCsoliwyDSariAmI3y0nyuCw.

Comments and suggestions from Reviewer 2 with authors’ responses

This paper investigates methylation levels within the ESR1 promoter in a cohort of female >40 years. Although there are some novel and interesting aspects to this work. I have a number of reservations about the work that necessitates addressing before this manuscript being in an acceptable state for publication in an academic journal.

  • “Although some of these modifications may be a random by-product of the aging process itself, others may promote beneficial adaptive changes.” Are the beneficial adaptive changes a product of natural selection?

Response: Associations between DNA methylation and adaptive evolution have not been clearly elucidated. Indeed, DNA methylation marks contribute to adaptive phenotypic variation, but, in mammals, they are erased during early development, following fertilization (Flores et al., 2013). It has been hypothesized that natural selection can alter environmentally-induced methylation of DNA by acting on the molecular mechanisms used for the genomic targeting of DNA methylation (Flores et al., 2013). For example, beneficial environmentally-induced methylation profiles (i.e.: changes promoting reproductive functions and longevity) may be maintained across generations through the selection of genomic mechanisms linked to these methylation profiles (Tikhodeyev, 2020). The hypothesis that the beneficial adaptive changes are a product of natural selection has been mentioned in the introduction (lns. 34-41, p.2).

  • “For instance, an epigenome-wide analysis revealed that age-associated DNA hypomethylation of distal regulatory elements (enhancers) was related to the upregulation of genes essential for cell identity and function”. Are these changes due to ageing or are they a by-product of it?

Response: Methylation changes due to aging may be described as adaptive (“epigenetic clock”) or random (“epigenetic drift”). This distinction is well described by Jones et al., (2015). “Epigenetic drift” may occur due to the loss of regulatory control with age, resulting in increased variability of DNA methylation across aging individuals. According to Jones et al. (2015), epigenetic drift may be defined as a by-product of the aging process itself. Instead, the “epigenetic clock” leads to common DNA methylation changes across aging individuals. Some of these common changes may be beneficial for preserving health during aging (Jones et al., 2015). Relying on the conceptual distinction proposed by Jones et al., (2015), in the sentence “For instance, an epigenome-wide analysis revealed that age-associated DNA hypomethylation of distal regulatory elements (enhancers) was related to the upregulation of genes essential for cell identity and function”, we refer to the beneficial DNA methylation changes due to aging, and not to random by-products of it. We added this conceptual distinction with related explanations in the introduction (lns. 28-34, p. 2 and 42-43, p.2).

  • This is a little vague “All women were Caucasian, and most were Swiss or from neighboring countries”.

Response: We added information and changed this sentence as follows (lns. 9-11, p. 3): “All women were Caucasian, most were originally from Switzerland (89%) and the remaining part (11%) from the neighbouring German-speaking countries, Germany, Austria, and Liechtenstein.”

  • In what was is this relevant to the study? “and had a college/university vocational education (79.2%)”. What is relevant to a study of this nature is their nutritional status. For example, it would be worthwhile knowing about intakes of vitamin B12 and folate in a population such as this. In addition, it would be interesting to know a little about blood marker status and folate metabolism in these subjects e.g. such as homocysteine levels, B12, red cell folate.

Response: Women participating in our study had vocational secondary education or higher education, and the level of education was not associated with ER gene methylation (r= -0.007; p= 0.937). However, educational levels have been reported to correlate with DNA methylation in two previous studies (Dongen et al., 2018; Tehranifar et al., 2013), as well as with health in advanced age (Joung et al., 2000). Therefore, the information on educational level may be used for the comparison of studies’ populations. Unfortunately, in this study, we didn’t analyse blood markers linked to women’s nutritional status, such as homocysteine levels, B12, or red cell folate. This point has been added as a study limitation, as follows (lns 46-47, p.6): “. At last, in this study, we did not assess data linked to women’s nutritional status, such as B12 and red cell folate, which have been shown to regulate DNA methylation (Mahajan et al., 2019)”.

  • In table one it would be worthwhile including the median age in this cohort.

Response: The median age has been added in Table 1 (ln. 16, p. 3).

  • Figure one B why is there no r value on the correlation. Is there a trend. A Persons correlation would be useful.

Response: Pearson correlation (r= 0.224, p= 0.014) has been added to Figure 1B and in the figure’s caption (ln. 3, p. 4).

  • “In this study, we explored DNA methylation of key regulatory regions of ESR1 in association with menopausal status, age, and E2 levels in healthy middle-aged and older women. Methylation levels were low at promoters A and B, while intermediate methylation levels were found at CpGI shore. Postmenopause and lower E2 levels were associated with lower methylation of CpGI shore, and the effect of E2 levels was significant also after adjusting for menopausal status and age. This association was stronger at CpG 3 (Illumina probe: cg07746998), CpG 9 (Illumina probe: cg17264271), and CpG 1 (not included among Illumina probes). CpGI shore methylation was positively associated with E2 levels in pre- and postmenopausal women, while this association was negative in perimenopausal women. Postmenopause was associated with increased methylation of promoter B30 at CpG 7 (Illumina probe: cg22839866), while age was positively associated with methylation of promoter B at CpG 12 (Illumina probe: cg13612689) after adjusting for E2 levels and menopausal status. Concerning methylation of promoter A, we didn’t observe association with menopausal status, E2 levels, or age.”

If we make the assumption that these findings are robust. What can be concluded about the relationship between ageing, the menopause and DNA methylation from an evolutionary context?

Response: DNA methylation changes that reflect a programmed process are perhaps selected through evolution (Field et al., 2018). There is increasing evidence indicating that the hypomethylation of enhancers may be an example of these programmed DNA methylation changes (Field et al., 2018). For instance, hypomethylation of enhancers has been implicated as a component of the mouse clock (Field et al., 2018), and has been shown to preserve cell-functions in mouse pancreatic β-cells during aging (Avrahami et al., 2015). Hormonal changes related to menopause have been described as potential determinants of the epigenetic clock (Jylhävä et al., 2017). In this study, we described an association between E2 levels and ERα enhancer methylation that may promote healthy aging. In an evolutionary context, this hormone-DNA methylation association would have positive effects on fitness-related traits earlier in life, as the strength of natural selection decreases with age (Fabian and Flatt, 2011). These remarks have been added to the “Discussions” section (lns. 44-51, p. 5 and lns. 1-2, p. 6).

Minor changes

Response: All changes have been implemented at the indicated lines and pages.

  • Please change “to maintain” to “ to maintaining” (ln. 52, p. 2).
  • Change “ESR1 promoter” to “ the ESR1 promoter”(ln. 2, p. 3).
  • Change “insight for further investigations in the field of women healthy aging.” to “insights for further investigations in the field of female healthspan” (ln. 5, p. 3).
  • Please change “Concerning methylation of promoter A, we didn’t observe association with menopausal status, E2 levels or age.” To “Concerning the methylation of promoter A, we didn’t observe an association with menopausal status, E2 levels or age (ln. 15, p. 5).
  • Please change “Our findings on ESR1 methylation associated with E2 levels and age present similarities with previous researches.” To “Our findings on ESR1 methylation associated with E2 levels and age present similarities with previous research” (ln. 18, p. 5).
  • This needs to be phrased better “Therefore, our sample was representative of the healthy population and major illnesses couldn’t have biased the results.” To “Therefore, the results could not have been biased by major illnesses” (lns. 27-28, p.6).
  • Please change “Although there is evidence indicating that increased CpGI shore methylation is associated to decreased ESR1 expression in various tissues.” To “Although there is evidence indicating that increased CpGI shore methylation is associated with decreased ESR1 expression in various tissues” (ln. 29, p. 6).
  • I don’t know if it is a journal requirement but presently where the conclusions are located in the manuscript is not appropriate. Conclusions need to be after the discussion.

Response: The location of the conclusions is a journal requirement.

We hope that our revision meets your approval. We would be glad to respond to any further questions and comments that you may have.

Yours sincerely,

Ulrike Ehlert 

References

Avrahami, D., Li, C., Zhang, J., Schug, J., Avrahami, R., Rao, S., … Kaestner, K. H. (2015). Aging-Dependent Demethylation of Regulatory Elements Correlates with Chromatin State and Improved β Cell Function. Cell Metabolism, 22(4), 619–632. https://doi.org/10.1016/j.cmet.2015.07.025

Fabian, D., & Flatt, T. (2011). The evolution of aging. Nature Education Knowledge, 3(3), 1–10.

Field, A. E., Robertson, N. A., Wang, T., Havas, A., Ideker, T., & Adams, P. D. (2018). DNA Methylation Clocks in Aging: Categories, Causes, and Consequences. Molecular Cell, 71(6), 882–895. https://doi.org/10.1016/j.molcel.2018.08.008

Flores, K. B., Wolschin, F., & Amdam, G. V. (2013). The role of methylation of DNA in environmental adaptation. Oxford University Press.

Joung, I. M. A., Kunst, A. E., van Imhoff, E., & Mackenbach, J. P. (2000). Education, aging, and health: to what extent can the rise in educational level relieve the future health (care) burden associated with population aging in the Netherlands? Journal of Clinical Epidemiology, 53(9), 955–963.

Jylhävä, J., Pedersen, N. L., & Hägg, S. (2017). Biological age predictors. EBioMedicine, 21, 29–36.

Mahajan, A., Sapehia, D., Thakur, S., Mohanraj, P. S., Bagga, R., & Kaur, J. (2019). Effect of imbalance in folate and vitamin B12 in maternal/parental diet on global methylation and regulatory miRNAs. Scientific Reports, 9(1), 17602. https://doi.org/10.1038/s41598-019-54070-9

Tehranifar, P., Wu, H.-C., Fan, X., Flom, J. D., Ferris, J. S., Cho, Y. H., … Terry, M. B. (2013). Early life socioeconomic factors and genomic DNA methylation in mid-life. Epigenetics, 8(1), 23–27.

Tikhodeyev, O. N. (2020). Heredity determined by the environment: Lamarckian ideas in modern molecular biology. Science of The Total Environment, 710, 135521.

van Dongen, J., Bonder, M. J., Dekkers, K. F., Nivard, M. G., van Iterson, M., Willemsen, G., … Franke, L. (2018). DNA methylation signatures of educational attainment. Npj Science of Learning, 3(1), 1–14.